

# The boring monopole

**Prateek Agrawal and Michael Nee⋆**

Rudolf Peierls Centre for Theoretical Physics, University of Oxford,
Parks Road, Oxford OX1 3PU, United Kingdom

⋆ michael.nee@physics.ox.ac.uk

## Abstract

We study false vacuum decays catalysed by metastable magnetic monopoles which act as tunnelling sites with exponentially enhanced decay rates. Metastable monopoles are configurations where the monopole core is in the true vacuum of the scalar potential. The field profiles describing the decay do not have the typically assumed $O(3)/O(4)$ symmetry, thus requiring an extension of the usual decay rate calculation. To numerically determine the saddle point solutions which describe the tunnelling process we use a new algorithm based on the mountain pass theorem. This method can be applied more widely to phase transitions with reduced symmetry, such as decays away from the zero and infinite temperature limits. In our setup monopole-catalysed tunnelling can dominate over the homogeneous false vacuum decay for a wide range of parameters, significantly modify the gravitational wave signal or trigger phase transitions which would otherwise never complete. A single boring monopole in our Hubble patch may determine the lifetime of our current vacuum.

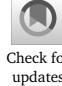

# 1 Introduction

First order phase transitions occur ubiquitously in nature, from the boiling phase transition of water to bubble formation in cloud chambers. These transitions are often strongly catalysed by boundary effects or defects which exponentially dominate the rate of transition. It is therefore natural to ask whether cosmological phase transitions may be catalysed by defects in the same way.

In many theories of physics beyond the Standard Model (BSM) first order phase transitions play an important role during the early universe. The rate at which the phase transition proceeds largely determines its phenomenological consequences. In theories of electroweak baryogenesis [1] the rate strongly influences the final baryon asymmetry, while in Randall-Sundrum models [2] the rate of the confining transition must be sufficiently rapid to avoid an eternally inflating phase [3], leading to strong bounds on the model. These first order phase transitions occur around the TeV scale and lead to a potentially observable gravitational wave signal with features which depend strongly on the phase transition rate.

In models where we live in a metastable universe, the stability of the vacuum is set by the rate of a first order phase transition to a lower energy vacuum state. This is realised in the picture of the string landscape where our vacuum, with a small cosmological constant, is anthropically selected from a large number of vacua. Even in the absence of additional sectors, the SM extended to very high energies has a second, lower energy vacuum at large values of the Higgs field [4–12]. In this work, we show that the decay rate of the metastable vacuum can be exponentially dominated by defects, even if the distribution of these defects is very sparse on cosmological scales. In such cases the lifetime of the universe is set by the lifetime of the defects.

We study models of vacuum decay catalysed by the 't Hooft–Polyakov monopole [13,14], with a scalar field potential such that the symmetry breaking vacuum is metastable and the symmetry preserving vacuum is a global minimum. In this case the monopole is metastable for a range of parameters and can tunnel to an expanding true vacuum bubble. Classically unstable monopoles were studied in [15,16] and metastable monopoles were studied in the thin-wall approximation in [17,18].

This idea can be generalised to other topological defects such as global monopoles, strings [19–21] and domain walls. The defect need not be topologically stable [22,23] and there exist studies of Q-balls [24–26] and black holes as nucleation sites [27–33] which can affect the electroweak lifetime [34–38]. The gravitational effect of compact objects was considered in [39] and finite density effects in [40]. In each of these studies the catalysed decay is analysed using a thin-wall approximation or in the limit of classically unstable defects, although this is far from the most general scenario.

The usual homogeneous false vacuum decay proceeds through bubble nucleation. The rate of transitions is calculated through semiclassical field configurations which are solutions to Euclidean field equations [41–43], also known as bounce solutions. The Euclidean action for the bounce configuration sets the exponent of the decay rate. The bounce action is usually calculated in the limit of zero or infinite temperature, where the field profiles have an $O(3)$ or $O(4)$ symmetric dependence on the coordinates.

The saddle point describing monopole decay does not have $O(3)$ or $O(4)$ symmetry in Euclidean space, and is difficult to find away from the thin-wall limit studied in [17, 18]. We present a new method to determine the solutions to catalysed tunnelling problems in general, using the mountain pass theorem (MPT) to numerically find the saddle point. This provides the first steps towards reliable calculations of the lifetime of metastable vacua when defects are present.

There are many applications of this calculation, which depend on the mechanism by which monopoles are populated. They could arise from a Kibble-Zurek mechanism from an earlier phase transition, be produced in high energy collisions or black hole evaporation. The production can leave an imprint on the gravitational wave background both through the rate of the phase transition, which determines the amplitude and peak frequency of the signal, and through the spatial distribution of magnetic monopoles which can lead to an observable angular anisotropy in the signal.

We review false vacuum decay and the physics of metastable monopoles in the thin-wall limit in section 2. We present our method to evaluate the bounce action in section 3. We present some applications in section 4 and conclude in section 5.

## 2  Monopoles & Vacuum Decay

In this section we review the formalism for calculating the lifetime of a homogeneous false vacuum state and compare this to vacuum decay seeded by magnetic monopoles in the thin wall limit. The magnetic charge of the monopole may correspond to the SM $U(1)_{\text{em}}$ or to a hidden sector. For concreteness we consider a model with an $SU(2)$ gauge symmetry, and a scalar field $\phi$ which is charged in the triplet representation. It is possible that the monopoles come from the breaking of a larger gauge group, in which case $SU(2)$ represents a subgroup of the full gauge symmetry. The scalar potential we consider is

$$V_\phi(\phi^\dagger\phi) = \lambda v^4\left(\frac{1}{2}\frac{(\phi^\dagger\phi)}{v^2} - \left(1 - \frac{3\epsilon}{\lambda}\right)\frac{(\phi^\dagger\phi)^2}{v^4} + \frac{1}{2}\left(1 - \frac{4\epsilon}{\lambda}\right)\frac{(\phi^\dagger\phi)^3}{v^6}\right), \tag{1}$$

where $\lambda, \epsilon$ are dimensionless parameters. The shape of the potential is shown in figure 1. This form of the potential is chosen so that for $0 < \epsilon < \lambda/6$, $\phi = 0$ is the true minimum of the potential, and there is a second set of (metastable) minima at $|\phi| = v$ separated from the true vacuum by a potential barrier. $\epsilon$ parametrises the difference in potential energy between the vacua:

$$V_\phi(v) - V_\phi(0) = \epsilon v^4, \tag{2}$$

and $\lambda$ sets the overall size of the potential.

The gauge sector is not relevant for the homogeneous false vacuum tunnelling process, and we choose this specific set up anticipating the discussion of magnetic monopoles catalysing the phase transition in section 2.2. Since $\phi$ is charged under $SU(2)$, terms in the potential must be powers of the gauge invariant combination $\phi^\dagger\phi$, so in order to satisfy both these conditions the potential must be stabilised by an operator with dimension $> 4$. We are assuming that

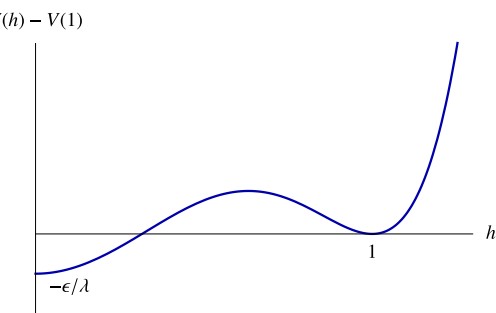

Figure 1: The scalar potential for the metastable monopole model, given a representative set of values.

the $(\phi^\dagger\phi)^3$ stabilises the potential at large $\phi$ and we can ignore operators with dimension > 6, although adding higher order terms with positive coefficients does not alter our results significantly.

The tunnelling process involve bubble profiles that are spatially spherically symmetric. It will be convenient to parametrise the $SU(2)$ triplet $\phi$ by a dimensionless profile $h$ defined by

$$\phi^\dagger\phi = v^2 h^2. \tag{3}$$

For the homogeneous false vacuum decay the orientation of the scalar $\phi$ is arbitrary and constant, whereas for the monopole solution it is given by the hedgehog configuration (equation 13).

The tunnelling solutions will be dictated by the scalar field profile $h$. We will use a subscript 'fvb' to refer to the false vacuum bounce solution, subscript 'm' to the static monopole profile and 'mb' to the monopole tunnelling solution. We will often make use of the rescaled potential $V$ written in terms of $h$ with the parameter $\lambda$ factored out:

$$\lambda v^4 V(h) = V_\phi(v^2 h^2). \tag{4}$$

In both cases we work in the zero temperature limit and assume that the effects of gravity are negligible.

## 2.1 False vacuum decay

The rate for the false vacuum tunnelling process, defined as the rate of bubble nucleation per unit volume is given by [42, 43]

$$\Gamma_{\text{fvb}} = A^4 e^{-B_{\text{fvb}}}. \tag{5}$$

The dominant suppression of the decay rate comes from the exponential dependence on the bounce action $B_{\text{fvb}}$, and the calculation of this quantity is the focus of this paper. In equation (5) $A$ is related to the determinant of fluctuations around the bounce solution [43], and only corrects the decay rate by logarithmic correction to $B_{\text{fvb}}$. Throughout this paper we will simply approximate $A \sim v$. $B_{\text{fvb}}$ is given by the Euclidean action $S_E$ evaluated on the bounce solution $|\phi| = v h_{\text{fvb}}$

$$B_{\text{fvb}} = S_E[v h_{\text{fvb}}] - S_E[v], \tag{6}$$

where for our case of a single scalar field the action is given by

$$S_E[\phi] = \int dt_E d^3x \left[ \frac{1}{2}|\partial\phi|^2 + V(\phi) \right]. \tag{7}$$

The bounce profile $h_{\text{fvb}}(t_E, \vec{x})$ is a solution to the Euclidean field equations and describes a field configuration where $h_{\text{fvb}}$ begins in the false vacuum state ($h_{\text{fvb}} = 1$) at $t_E \to -\infty$, evolves to a bubble of true vacuum before turning around and evolving along a time-reversed path to the false vacuum state [42]. The translational symmetry of the field equations allows us to set the turning point of the solution to $t_E = 0$. It can be shown that in the zero temperature limit the solution obeys an $O(4)$ symmetry and depends only on the combination $\rho = \sqrt{t_E^2 + r^2}$ [44]. The field equations then become

$$h_{\text{fvb}}'' + \frac{3}{\rho} h_{\text{fvb}}' = \lambda v^2 \frac{\partial V(h_{\text{fvb}})}{\partial h}, \tag{8}$$

with boundary conditions

$$h_{\text{fvb}}(\rho \to \infty) = 1, \qquad\qquad h_{\text{fvb}}'(0) = 0, \tag{9}$$

where $'$ denotes the derivate $\partial/\partial\rho$.

In general the solution $h_{\text{fvb}}$ can only be found by numerically integrating equation (8), however in the case where the two vacua are nearly degenerate a simplifying approximation can be made. In this case the solution remains near the true vacuum at $h = 0$ until $\rho$ is large enough so that the friction term ($\propto h_{\text{fvb}}'$) is negligible before rapidly evolving to the false vacuum value $h_{\text{fvb}} = 1$. In this case the profile is given by the thin-wall approximation:

$$h_{\text{fvb}} = \begin{cases} 0, & \rho < R - \delta, \\ \bar{h}_{\text{fvb}}(\rho), & |\rho - R| < \delta, \\ 1, & \rho > R + \delta, \end{cases} \tag{10}$$

where $\bar{h}_{\text{fvb}}(\rho)$ interpolates between 0 and 1 in a small interval with length $2\delta \ll R$ around $R$.

The energy of a static bubble of radius $R$, relative to that of the false vacuum is given by

$$E_{\text{fvb}}(R) = -\frac{4\pi\epsilon v^4}{3} R^3 + 4\pi\sigma R^2, \tag{11}$$

where $\sigma$ is the surface tension

$$\sigma = \sqrt{2\lambda} v^3 \int_0^1 dh \sqrt{V(h) - V(0)}. \tag{12}$$

$E_{\text{fvb}}$ has a maximum at $R_c = 2\sigma/(\epsilon v^4)$ and any bubble of true vacuum larger than this is unstable against expanding, as shown in figure 2. As $R = 0$ corresponds to the homogeneous false vacuum ($\phi = v$ everywhere) and energy must be conserved, the nucleated bubble will have radius $R_{\text{fvb}} = 3\sigma/(\epsilon v^4)$. A bubble nucleated at this radius will spontaneously expand with the latent heat released from converting a region of false vacuum to true vacuum driving the expansion of the bubble wall. This picture also highlights that the bounce solution is a saddle point of $S_E$, with a single negative mode corresponding to the expansion of the bubble.

The thin-wall limit, while instructive, is not generally applicable and applies only in the regime where the transition rate is most suppressed (nearly degenerate vacua separated by a large potential barrier). In general solutions to equation (8) have to be found numerically. A simple method to do this is using a shooting algorithm. The algorithm works by making a guess $h_0$ for the value of $h_{\text{fvb}}$ at $\rho = 0$, then solving equation (8) as an initial value problem with boundary conditions $h_{\text{fvb}}(0) = h_0$, $h_{\text{fvb}}'(0) = 0$. If the guess $h_0$ is too close to the true vacuum at $h = 0$ the solution found will "overshoot", evolving past the false vacuum $h = 1$ and diverging toward $+\infty$ at large $\rho$, while if the guess is too far from the true vacuum the solution will "undershoot" and reach a maximum value less than 1 before turning around and oscillating around the field value that maximises $V(\phi)$. The bounce solution separates the profiles that overshoot from those that undershoot.

## 2.2 Metastable monopole background

We now turn to the case where false vacuum decay can be seeded by magnetic monopoles, which act as impurity sites and catalyse the phase transition from the false to true vacuum. This effect is relevant when:

1. the scalar field $\phi$ is charged under a gauge group $G$ and the false vacuum state breaks the gauge group $G$ to a subgroup $H$,

2. the true vacuum state is the symmetry restored phase, $\phi = 0$,[1] and

3. the symmetry breaking pattern $G \to H$ leads to topological monopole solutions.

The symmetry breaking pattern ($G$ spontaneously broken to a subgroup $H$) permits topological monopole solutions if the homotopy group $\pi_2(G/H)$ is non-trivial. Our setup, described above equation (1) is the simplest example where these conditions are satisfied. In this case the field profiles describing the monopole solution take the form

$$
\begin{aligned}
\phi^a &= v \hat{r}^a h_{\mathrm{m}}(r), \\
A_i^a &= \epsilon^{aij} \hat{r}^j \left( \frac{1 - u_{\mathrm{m}}(r)}{g r} \right), \\
A_0^a &= 0,
\end{aligned}
\tag{13}
$$

where $g$ is the gauge coupling, $a$ indexes the adjoint representation of $SU(2)$ and $i, j$ are spatial indices. At large values of $r$ the scalar field must approach a point on the symmetry breaking minimum of the potential, $\phi^\dagger \phi = v^2$, which implies $h_{\mathrm{m}} \to 1$ as $r \to \infty$; while the gauge field profile $u_{\mathrm{m}}$ approaches 0 at large $r$, giving the coulomb field for a monopole with magnetic charge $Q_{\mathrm{m}} = 4\pi/g$. As each point on the spatial unit sphere is mapped to a distinct orientation in field space (as $\phi^a \propto \hat{r}^a$) the only way for this solution to be continuous as $r \to 0$ is if $h_{\mathrm{m}}(0) = 0$. The topology of the gauge field profile similarly requires that $u_{\mathrm{m}}(0) = 1$. By construction $h = 0$ is the true minimum of the potential $V$, so in some neighbourhood around the centre of the monopole the scalar field is in the true vacuum region of the potential. In this way, the monopole profile resembles the profile of a bubble nucleated in false vacuum decay.

When these assumptions hold there are now two separate processes that contribute to the decay of the false vacuum. The formalism described in section 2.1 is still valid when considering bubbles nucleated far from the core of a monopole, but around the core of a monopole the rate of bubble nucleation can be exponentially enhanced, as the interior of the monopole contains a region of true vacuum. If this interior region of the monopole is of a critical size, then the monopole can become classically unstable and will spontaneously expand – even if the false vacuum decay process is exponentially suppressed. If the monopole is smaller than the critical size, it can still decay via a tunnelling process where the monopole tunnels to a larger monopole before spontaneously expanding.

Similar to the false vacuum tunnelling process, the rate for this process will be dictated by bounce solutions to the Euclidean field equations. The exponent suppressing the monopole decay, $B_{\mathrm{mb}}$, is the Euclidean action evaluated on these solutions. The decay rate of a single monopole is

$$
\Gamma_{\text{single-monopole}} = A' e^{-B_{\mathrm{mb}}}.
\tag{14}
$$

---

[1]This assumption can be replaced by the less restrictive requirement that $\phi = 0$ is in the basin of attraction of the true vacuum, but we consider the case where $\phi = 0$ is precisely at the true vacuum both for simplicity and as this assumption maximises the effect under consideration.

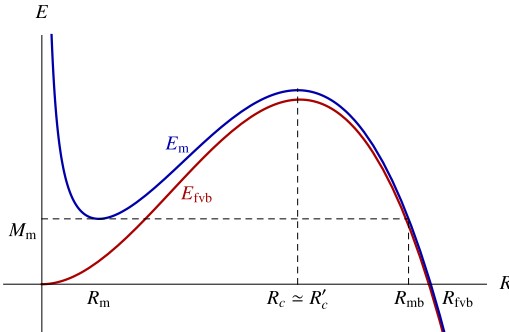

Figure 2: Energy of the metastable monopole (blue) and false vacuum bubble (red) as a function of radius in the thin-wall limit. We neglect differences in the tension of the two bubbles and set $\sigma = \sigma'$. The plot is illustrative of the shape of profiles in the thin-wall limit.

The contribution of the monopole channel to the false vacuum decay rate (per unit volume) in this case will then be given by

$$\Gamma_{\text{mb}} = n_{\text{m}} A' e^{-B_{\text{mb}}}, \tag{15}$$

where $n_{\text{m}}$ is the monopole number density. In general the pre-factors $A$ in equation (5) and $A'$ may be different, but we make the approximation that $A \sim A' \sim v$, which only affects the bounce action logarithmically. Due to the exponential dependence on $B_{\text{fvb}}, B_{\text{mb}}$ in equations (5) and (15) if $B_{\text{mb}} < B_{\text{fvb}}$ the monopole-catalysed decay can still dominate the false vacuum process even if $n_{\text{m}}$ is very small.

## 2.3 Metastable monopoles in the thin wall limit

In this section we discuss the behaviour of metastable monopoles in the thin wall limit [18] and compare this to the false vacuum tunnelling process. This amounts to making a similar approximation to the one in equation (10), where the scalar field interpolates between the two vacua in a small region around the radius $R$. The detailed form of the profile $u_{\text{m}}$ is not important at leading order, and it is assumed that $u_{\text{m}}$ varies with the same characteristic radius as the scalar field [18]. In this limit the energy of the monopole can be written solely as a function of the radius $R$ as

$$E_{\text{m}}(R) = -\frac{4\pi\epsilon v^4}{3} R^3 + 4\pi\sigma' R^2 + \frac{2\pi}{g^2 R}, \tag{16}$$

where $\sigma'$ is the surface tension of the monopole. $\sigma'$ is in general different from the surface tension $\sigma$ defined in equation (11) due to contributions from the gauge field profile, but these become less important at larger $R$ values. For small $\epsilon$ the energy has a minimum at $R_{\text{m}} \simeq \left(4g^2\sigma'\right)^{-1/3}$ and a maximum at $R_c \simeq 2\sigma'/(\epsilon v^4)$. Comparing this with the energy (10) of a bubble nucleated in a false vacuum tunnelling process we see there is a minimum generated at a small radius in the monopole case due to the additional term $2\pi/(g^2 R)$, as shown in figure 2. The monopole can thus be thought of as as a sub-critical bubble of true vacuum which is stabilised by the magnetic self-energy in the thin-wall limit.[2] Therefore, monopoles do not collapse to zero size, whereas sub-critical false vacuum bubbles do.

---

[2]Note that global monopoles are also stable against collapse – in this case the outward pressure is provided by the gradient energy of the scalar field profile, which is fixed to vanish at the origin due to topology. However, there is no thin-wall limit for global monopoles.

Choosing parameters such that $R_m = R_c$ (i.e. if $g$ is sufficiently small) leads to a monopole which is classically unstable and will spontaneously expand as was the case discussed in ref.'s [15–17]. Even if $R_m < R_c$ the monopole can tunnel to a monopole solution of (super)-critical size ($R \geq R_c$) at which point it will spontaneously expand. In contrast, the false vacuum case can be thought of as tunnelling from the $R = 0$ homogeneous false vacuum to a bubble of size $R \geq R'_c$. It is therefore a natural expectation that the bounce action for monopole decay should be smaller than the false-vacuum bounce.

The monopole bounce action in the thin-wall approximation was studied in ref. [18], their result was:

$$\frac{B_{\mathrm{mb}}}{B_{\mathrm{fvb}}} = \frac{32\sqrt{2}}{105\pi} \left(1 - \frac{R_m}{R_{\mathrm{mb}}}\right)^{5/2} I\left(\frac{R_m}{R_{\mathrm{mb}}}\right), \tag{17}$$

where $I$ is an $\mathcal{O}(1)$ function. Here $R_{\mathrm{mb}}$ is the radius of the bubble after tunnelling, given by $E_{\mathrm{m}}(R_{\mathrm{mb}}) = E_{\mathrm{m}}(R_m)$ as required by energy conservation. This expression makes explicit the classical instability as $R_m \to R_{\mathrm{mb}}$ and the barrier in $E_{\mathrm{m}}(R)$ disappears.

## 2.4 Thin wall limitations

The thin wall limit only gives a good approximation to the true monopole solution in a limited set of circumstances. In general, the field profiles $h_{\mathrm{m}}, u_{\mathrm{m}}$ have a thick-walled profile which is determined by the field equations

$$
\begin{aligned}
h_{\mathrm{m}}'' + \frac{2}{s} h_{\mathrm{m}}' &= \frac{2 h_{\mathrm{m}} u_{\mathrm{m}}^2}{s^2} + \frac{\lambda}{g^2} \frac{\partial V(h_{\mathrm{m}})}{\partial h}, \\
u_{\mathrm{m}}'' &= \frac{u_{\mathrm{m}}(u_{\mathrm{m}}^2 - 1))}{s^2} + h_{\mathrm{m}}^2 u_{\mathrm{m}},
\end{aligned}
\tag{18}
$$

where $s = gvr$ is a rescaled radial coordinate.

The usual justification for the thin-wall approximation is that an equation of motion of the form of equation (8) can be thought of as describing a particle (with position $\varphi$, 'time' coordinate $\rho$) moving on an inverted potential $-V(\varphi)$ with a $\rho$-dependent friction term. If the minima of $V$ are nearly degenerate (i.e. if $\epsilon \ll 1$) the particle must remain at the true vacuum point of the potential until the friction term becomes negligible before moving rapidly to the false vacuum minimum of the potential. Comparing equation (18) to (8), it is clear that this logic does not apply straightforwardly for the monopole equation of motion due to presence of the interaction term $2hu^2/s^2$. In particular, if we take the limit that the interaction dominates the term from the scalar potential, $\lambda/g^2 \ll 1$ (known as the BPS limit [45]), then the exact solution is known to be:

$$h_{\mathrm{bps}}(s) = \coth(s) - \frac{1}{s}, \quad u_{\mathrm{bps}}(s) = \frac{s}{\sinh(s)}, \tag{19}$$

and does not describe a thin-wall profile, regardless of the value of $\epsilon$. We therefore expect that a thin wall approximation for the scalar field profile is only valid for both $\lambda \gg g^2$ and $\epsilon \ll 1$, while more general monopoles will have a thick-walled profile.

At distances $s \lesssim g\lambda^{-1/2}$ from the centre of the monopole the potential term is always negligible in comparison to the interaction term. Expanding equation (18) for small $s$ the leading term for $h$ is linear in $s$, and this coefficient is only non-zero due to the interaction term in the equation. This is in conflict with the assumption that the scalar field stays exponentially close to the true vacuum value inside the thin wall bubble, which is crucial to deriving the first term of equation (16). Even for small deviations of $h$ from zero, $V(h)$ can become positive since the thin wall limit requires $|V(0)| = \epsilon \ll 1$. As the EOM for $u_{\mathrm{m}}$ contains no additional parameters,

we similarly expect that $u_{\mathrm{m}}$ evolves smoothly between 1 and 0 as $s$ varies by an $\mathcal{O}(1)$ amount, although this does not significantly affect the validity of the thin wall approximation of the bounce action.

While the thin wall approximation for the monopole profile is only expected to hold for a specific set of parameters we expect that the intuitive picture derived from this approximation should apply more generally. A thick-walled monopole must have a region at its core which is in the true vacuum region of the potential and therefore we expect that a bubble nucleated around a monopole should have a smaller bounce action than a bubble nucleated from the homogenous false vacuum. This motivates looking at more general monopole profiles to determine the bounce action which describes their decay, although this is complicated by the fact that they can no longer be described solely by their radius. The calculation of this process in generality is the main result of this work, which we present in the next section.

## 3 General Monopole Tunnelling

In this section we describe a numerical method for calculating the bounce action for the tunnelling process initiated by a monopole. The bounce action for this process is given by:

$$B_{\mathrm{mb}} = S_E[h_{\mathrm{mb}}, u_{\mathrm{mb}}] - S_E[h_{\mathrm{m}}, u_{\mathrm{m}}], \qquad (20)$$

where $S_E$ is the Euclidean action for the gauge field and triplet scalar, and $h_{\mathrm{m}}, u_{\mathrm{m}}$ are the monopole field profiles obtained from solving equation (18).

The functions $h_{\mathrm{mb}}, u_{\mathrm{mb}}$ describe the tunnelling process from the monopole solution to a critical bubble of true vacuum and will depend on the variable $s$ and the (rescaled) Euclidean time $\tau = g v t_E$, assuming the monopole bounce solutions retain the $O(3)$ symmetric dependence on the spatial coordinates. They solve the equations of motion

$$
\begin{aligned}
\ddot{h}_{\mathrm{mb}} + h''_{\mathrm{mb}} + \frac{2}{s} h'_{\mathrm{mb}} &= \frac{2 h_{\mathrm{mb}} u_{\mathrm{mb}}^2}{s^2} + \frac{\lambda}{g^2} \frac{\partial V(h_{\mathrm{mb}})}{\partial h}, \\
\ddot{u}_{\mathrm{mb}} + u''_{\mathrm{mb}} &= \frac{u_{\mathrm{mb}}(u_{\mathrm{mb}}^2 - 1)}{s^2} + h_{\mathrm{mb}}^2 u_{\mathrm{mb}},
\end{aligned}
\qquad (21)
$$

where a $\dot{}$ denotes $\partial/\partial\tau$ and $'$ denotes $\partial/\partial s$. There are two boundary conditions enforced by the topology of the profiles, $h_{\mathrm{mb}}(\tau, 0) = 0$ and $u_{\mathrm{mb}}(\tau, 0) = 1$, and in order to have a finite tunnelling action we require that $h_{\mathrm{mb}}, u_{\mathrm{mb}}$ approach the monopole solutions $h_{\mathrm{m}}(s), u_{\mathrm{m}}(s)$ as $|\tau| \to \infty$. Combined with the symmetry of the equations under $\tau \to -\tau$ this implies that for some $\tau$ the derivatives $\dot{h}_{\mathrm{mb}}, \dot{u}_{\mathrm{mb}}$ vanish. As the equations are invariant under shifts in $\tau$ we can choose this point to be $\tau = 0$, giving us the final boundary condition $\dot{h}_{\mathrm{mb}}(0, s) = \dot{u}_{\mathrm{mb}}(0, s) = 0$.

### 3.1 Mountain pass theorem

The system of equations (21) does not allow us to make the simplifications usually made when calculating the analogous expressions for tunnelling from the homogeneous false vacuum. Due to the explicit dependence on $s$ on the right hand side of equation (21), and the $s$-dependence of the boundary profiles $h_{\mathrm{m}}(s), u_{\mathrm{m}}(s)$ the solutions will no longer obey an $O(4)$ symmetry and will instead depend independently on $\tau$ and $s$. In figure 3 we show contours of the profiles $h_{\mathrm{m}}$, the $O(4)$ symmetric false vacuum bounce $h_{\mathrm{fvb}}$ and the monopole bounce $h_{\mathrm{mb}}$ that separate the false vacuum region from the true vacuum region in each case. Comparing the monopole and false vacuum bounce profiles it is clear that $O(4)$ symmetry is not a good approximation for the monopole bounce. The monopole bounce $h_{\mathrm{mb}}$ and the supercritical bounce $h_{\mathrm{scb}}$ are discussed in detail below.

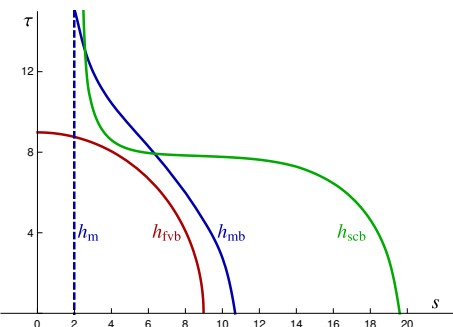

Figure 3: Lines in $\tau, s$ space which seperate the true vacuum and false vacuum regions for the $O(4)$ bounce solution (red), the monopole tunnelling solution (blue), the static monopole solution (blue dashed) and the supercritical profile (green). Parameters for each are $\lambda = 1/2$, $\epsilon = 5 \times 10^{-2}$, $g = 1$. To the left of each of the lines the relevant function is in the true vacuum region of the potential, while to the right they are on the false vacuum side of the potential.

This makes the system of equations (21) much more difficult to solve (at least in the absence of an approximation scheme such as the thin wall approximation). If we were to try to implement a shooting-type algorithm similar to the one described at the end of section 2.1 we would now need to guess the complete field profiles $h_{\mathrm{mb}}(0, s)$ and $u_{\mathrm{mb}}(0, s)$ for which the solutions asymptote to $h_{\mathrm{m}}(s)$ and $u_{\mathrm{m}}(s)$ as $\tau \to \infty$ after solving the equations of motion. This is impractical, both as it requires guessing the two complete field profiles and there is no way we know of to systematically improve the guess depending on the form of the asymptotic solutions found given an initial guess.

Instead, we use the fact that the solution we are looking for is a saddle point of $S_E$. If an initial guess which is close to the tunnelling solution is known in advance, it is possible to find the solution by varying the profiles so that they better satisfy the equations of motion [46]. However, this approach is limited by the need for a good initial guess, as there are other solutions to the field equations which the algorithm can relax to. Instead, we use a more generic algorithm which does not rely on the initial guess for convergence. It makes use of a theorem known as the mountain pass theorem (MPT) [47] to turn the problem of solving equations (21) into a minimisation problem, which can then be solved using a gradient descent algorithm. The use of the MPT allows us to find the bounce solutions without the requirement of a good initial guess for the profiles. The MPT is well-known, but its application to tunnelling in quantum field theory is novel.

The MPT applies to a functional $I[f]$ which maps a Hilbert space $H$ to the reals and satisfies the following conditions:

- $I$ is differentiable and the derivative $I'$ is Lipschitz continuous on bounded subsets of $H$;

- $I$ satisfies the Palais–Smale compactness condition, which guarantees the existence of a saddle point;

- $\exists f_0 \in H$ such that $I[f_0] = 0$ and $f_0$ is a local minimum of $I$;

- $\exists f_1 \in H$ such that $I[f_1] < 0$.

Considering the set of continuous paths that interpolate between $f_0$ and the function $f_1$

$$P = \{\gamma : [0, 1] \to H \mid \gamma(0) = f_0, \gamma(1) = f_1\}, \tag{22}$$

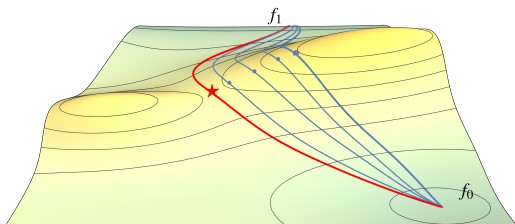

Figure 4: Mountain pass theorem in a two-dimensional example. Five paths interpolating between the minimum at $f_0$ and the negative point at $f_1$ are shown, where the red path passes through the saddle point (indicated by the red star). The blue dots show the maximum point along each of the other paths.

then the statement of the MPT is that the point

$$x = \min_{\gamma \in P} \max_{\alpha \in [0,1]} I[\gamma(\alpha)] \tag{23}$$

is a critical point of $I$. The intuition behind the theorem is that if we consider all paths which start at a minimum of $I$ where $I = 0$ and end at a point where $I < 0$, each path must reach a point where $I$ is maximum along the path (and $I$ is necessarily positive) and turn around. The path whose maximum point is smallest crosses through a saddle point of $I$ at this point. A simple example of such a set of paths is shown in figure 4. For each choice of path the maximisation picks out a point on the ridge, ensuring that the action is not lowered by exciting the negative mode. Therefore $\max_{\alpha \in [0,1]} I[\gamma(\alpha)]$ as a function of paths is robust against unstable modes and can be minimised using gradient descent methods.

The utility of the MPT for our purposes comes from realising that

$$I[h, u] = S_E[h, u] - S_E[h_{\mathrm{m}}, u_{\mathrm{m}}] \tag{24}$$

satisfies the conditions for the theorem to apply, and the solution $(h_{\mathrm{mb}}, u_{\mathrm{mb}})$ we are looking for is a saddle point of $I$. The monopole solution $(h_{\mathrm{m}}, u_{\mathrm{m}})$ for a metastable monopole is a local minimum of $I$ with $I[h_{\mathrm{m}}, u_{\mathrm{m}}] = 0$, and we can consider a field configuration describing a supercritical bubble $(h_{\mathrm{scb}}, u_{\mathrm{m}})$ which contains a large true vacuum region. A sufficiently large region of true vacuum will lead to $I[h_{\mathrm{scb}}, u_{\mathrm{m}}] < 0$, so the point $(h_{\mathrm{scb}}, u_{\mathrm{m}})$ in field space is the point $f_1$. If we consider paths which interpolate between $(h_{\mathrm{m}}, u_{\mathrm{m}})$ and $(h_{\mathrm{scb}}, u_{\mathrm{m}})$ then minimise the maximum value of $I$ along this set of paths we will approach the saddle point solution $(h_{\mathrm{mb}}, u_{\mathrm{mb}})$.

## 3.2 Mountain pass algorithm

Given the parametrisation of the scalar potential in equation (1) the solution for the monopole bounce $B_{\mathrm{mb}}$ depends on three parameters: $\lambda, g, \epsilon$. For a given set of these parameters we first find the monopole solutions $h_{\mathrm{m}}(s), u_{\mathrm{m}}(s)$ by solving equation (18) using a modified shooting algorithm described in Appendix A. We then construct the solution $h_{\mathrm{scb}}$ by using a convenient interpolation between the monopole profile and a supercritical bubble:

$$h_{\mathrm{scb}}(\tau, s) = \left(1 - e^{-(\tau/T)^2}\right) h_{\mathrm{m}}(s) + e^{-(\tau/T)^2} h_{\mathrm{sc}}(s, S, \delta),$$

$$h_{\mathrm{sc}}(s, S, \delta) = \frac{1}{2}\left[1 + \tanh\left(\frac{s - S}{\delta}\right)\right]. \tag{25}$$

The parameters $T, S$, and $\delta$ are chosen such that $I[h_{\mathrm{scb}}, u_{\mathrm{m}}] < 0$. An example of one such field profile is shown in figure 3. $T$ sets the time scale over which the scalar field varies between



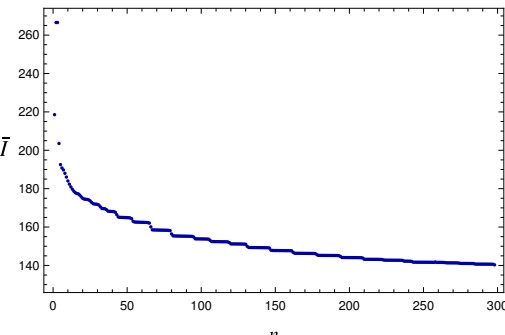
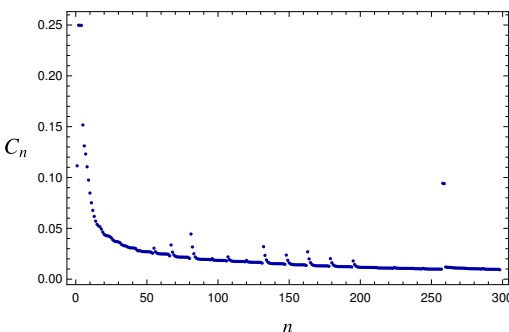

Figure 5: Value of the action $\bar{I}$ (left panel) and the cost function $C_n$ (right panel) for the field profiles $h_{\bar{\alpha}}, u_{\bar{\alpha}}$ as a function of the number iterations performed. Jumps in the values of $C_n$ are due to changes in the value of $\bar{\alpha}$ as the algorithm proceeds.

the bubble profile $h_{\mathrm{sc}}$ and the monopole solution $h_{\mathrm{m}}(s)$, $S$ sets the radius of the bubble profile and $\delta$ the thickness of the bubble. For $\delta \ll S$ this ansatz satisfies the boundary conditions for the equation of motion.

Having found the start and end points of the paths we want to consider, the next step is to choose a continuous interpolation between the monopole and supercritical solutions $h_\alpha(\tau, s), u_\alpha(\tau, s)$ for $0 \leq \alpha \leq 1$ such that $(h_0, u_0) = (h_{\mathrm{m}}, u_{\mathrm{m}})$ and $(h_1, u_1) = (h_{\mathrm{scb}}, u_{\mathrm{m}})$. This interpolation represents one choice of path between the endpoints, the task now is to find the path which passes through the saddle point of $I$. We present the algorithm here assuming $h_\alpha, u_\alpha$ are continuous in $\tau, s$ and $\alpha$, and give details on the discretised version in Appendix B. Having constructed the initial guess, the algorithm then proceeds as follows:

1. At a given step $n$ in the iteration calculate the functional

$$I[h_\alpha^n, u_\alpha^n] = S_E[h_\alpha^n, u_\alpha^n] - S_E[h_{\mathrm{m}}, u_{\mathrm{m}}], \qquad (26)$$

for $0 \leq \alpha \leq 1$. Let $\bar{\alpha}$ be the value for which $h_{\bar{\alpha}}^n, u_{\bar{\alpha}}^n$ maximise $I$ and let $\bar{I} = I[h_{\bar{\alpha}}^n, u_{\bar{\alpha}}^n]$ be this maximum value.

2. The functions $h_{\bar{\alpha}}^n, u_{\bar{\alpha}}^n$ are then updated to decrease the action $I[h_{\bar{\alpha}}^n, u_{\bar{\alpha}}^n]$ using a gradient descent algorithm

$$
\begin{aligned}
h_{\bar{\alpha}}^{n+1} &= h_{\bar{\alpha}}^n - \beta_n \frac{\delta \bar{I}}{\delta h_{\bar{\alpha}}^n}, \\
u_{\bar{\alpha}}^{n+1} &= u_{\bar{\alpha}}^n - \beta_n \frac{\delta \bar{I}}{\delta u_{\bar{\alpha}}^n},
\end{aligned}
\qquad (27)
$$

where $\beta_n$ is a step size, which we discuss in further detail in Appendix B.

Performing this step only on the functions $h_{\bar{\alpha}}^n, u_{\bar{\alpha}}^n$, however, will lead to $h_\alpha^{n+1}, u_\alpha^{n+1}$ becoming discontinuous in $\alpha$, violating a necessary condition on the choice of paths for the MPT to apply. To vary the functions while preserving continuity in $\alpha$ and keeping the endpoints at $\alpha = 0, 1$ fixed, we vary $h_\alpha^{n+1}, u_\alpha^{n+1}$ in the same direction as the change in $h_{\bar{\alpha}}^n, u_{\bar{\alpha}}^n$ but with a step size defined by a continuous function $F(\alpha, \bar{\alpha})$ such that $F(\bar{\alpha}, \bar{\alpha}) = 1, F(0, \bar{\alpha}) = F(1, \bar{\alpha}) = 0$, leading to

$$
\begin{aligned}
h_\alpha^{n+1} &= h_\alpha^n - \beta_n F(\alpha, \bar{\alpha}) \frac{\delta \bar{I}}{\delta h_{\bar{\alpha}}^n}, \\
u_\alpha^{n+1} &= u_\alpha^n - \beta_n F(\alpha, \bar{\alpha}) \frac{\delta \bar{I}}{\delta u_{\bar{\alpha}}^n}.
\end{aligned}
\qquad (28)
$$

We find that the specific choice of $F$ has little effect on the convergence of the algorithm provided the choice retains the required continuity in $\alpha$.

The variations of the field profiles at each step are simply the equations of motion evaluated on the field profiles $h_{\bar{\alpha}}^n, u_{\bar{\alpha}}^n$, albeit with a non-standard normalisation. Removing the indices $n, \bar{\alpha}$ for clarity, they are given by

$$
\begin{aligned}
\frac{\delta I}{\delta h} &= -\frac{8\pi s^2}{g^2}\left(\ddot{h} + h'' + \frac{2}{s}h' - \frac{2hu^2}{s^2} - \lambda\frac{\partial V(h)}{\partial h}\right), \\
\frac{\delta I}{\delta u} &= -\frac{16\pi}{g^2}\left(\ddot{u} + u'' - \frac{u(u^2-1)}{s^2} - h^2 u\right).
\end{aligned}
\tag{29}
$$

3. The magnitude of the change in field profiles is dependent on how well the equation of motion is satisfied at each point, and the algorithm will terminate when the equations of motion are satisfied.

In practice, we define a lattice of points $(\tau_j, s_i)$ and a cost function $C_n$ which gives a measure of how well the equation of motion is satisfied by the field profiles[3]

$$
C_n = \frac{1}{N_s N_\tau}\left[\sum_{ij}\left(\left(\frac{\delta\bar{I}}{\delta h_{\bar{\alpha}}^n}\right)_{ij}^2 + \left(\frac{\delta\bar{I}}{\delta u_{\bar{\alpha}}^n}\right)_{ij}^2\right)\right]^{1/2},
\tag{30}
$$

where $N_s, N_\tau$ are the number of points in $s, \tau$ that we sample and

$$
(h_{\bar{\alpha}}^n)_{ij} = h_{\bar{\alpha}}^n(\tau_j, s_i), \quad (u_{\bar{\alpha}}^n)_{ij} = u_{\bar{\alpha}}^n(\tau_j, s_i).
\tag{31}
$$

The cost function can be thought of as a measure of how well the field profiles $h_{\bar{\alpha}}^n, u_{\bar{\alpha}}^n$ satisfy the equations of motion.

The condition we use to terminate the algorithm is that $C_n < 10^{-2}$. As a final step the PDE is solved numerically for a perturbation around the solution found by the algorithm described above, but find that the perturbation found in this way amounts to a sub-percent level correction to the field profiles at each point. The choice of the termination condition is validated by requiring that the perturbations found by the PDE solver are small.

In figure 5 we show the action $\bar{I}$ and the cost $C_n$ as a function of the number of iterations $n$. In a gradient descent algorithm, the value of the action $\bar{I}$ is expected to decrease monotonically. This is not true for our implementation in general, as each iteration decreases $I[h_\alpha, u_\alpha]$ at a fixed value of $\alpha$. This might cause the value of $\alpha$ for which the updated profiles $(h_\alpha, u_\alpha)$ maximise $I$ to change, and possibly also cause $\bar{I}$ to increase. In such a case, the algorithm switches to updating the set of profiles at the new value of $\bar{\alpha}$ and proceeding with the gradient descent method. As can be seen from the right panel of figure 5, this behaviour does not prevent the algorithm from converging.

## 3.3 Numerical results

In this section, we present results comparing the monopole-catalysed and false vacuum decays in the simplified model described in section 2. We choose generic, $\mathcal{O}(1)$ values of the couplings $\lambda, g$ and take $\epsilon \lesssim \lambda/6$ as required for the potential to retain a metastable vacuum.

In the left panel of figure 6 we show the bounce action $B_{\mathrm{mb}}$ for tunnelling induced by a monopole in comparison to the false vacuum bounce $B_{\mathrm{fvb}}$ for different parameter values.

---

[3]For full details of the discretised algorithm we refer the reader to appendix B.

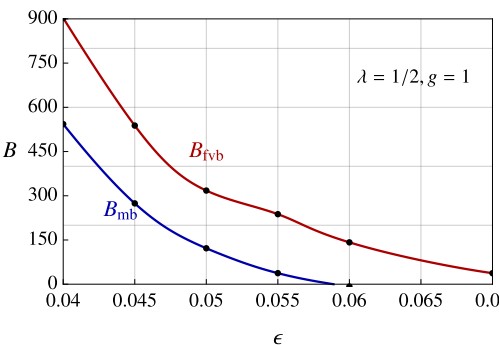
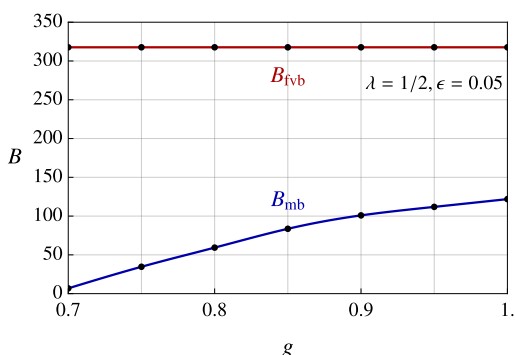

Figure 6: Plots showing the bounce action for bubble nucleation around a monopole ($B_{\mathrm{mb}}$) compared to the usual false vacuum bounce ($B_{\mathrm{fvb}}$). The left panel shows the two bounce actions as a function of $\epsilon$ for $\lambda = 1/2, g = 1$, the right panel shows the bounce actions as a function of $g$ for $\lambda = 1/2, \epsilon = 0.05$. The false vacuum bounce is independent of the gauge coupling so is constant in the right hand plot, showing that the monopole may be classically unstable (for $g < 0.7$ in our results) while the false vacuum decay is still exponentially suppressed.

The left panel of figure 6 shows the two bounce actions for $\lambda = 1/2, g = 1$ and varying $\epsilon$. The monopole bounce $B_{\mathrm{mb}}$ is always smaller than the false vacuum bounce and approaches a classical instability ($B_{\mathrm{mb}} \to 0$) at $\epsilon \sim 5.8 \times 10^{-2}$, where for the same choice of parameters $B_{\mathrm{fvb}} \sim 140$ so the false vacuum tunnelling rate is suppressed by a factor $\Gamma_{\mathrm{fvb}} \propto e^{-140}$.

The right panel of figure 6 shows the results for fixed $\lambda = 1/2$ and $\epsilon = 5 \times 10^{-2}$ and varying the coupling $g$. The false vacuum bounce is independent of the gauge coupling and is constant at $B_{\mathrm{fvb}} \sim 317$ while the monopole bounce decreases for smaller $g$, with the monopoles becoming classically unstable for $g \lesssim 0.7$. As the monopole size scales like $g^{-1}$, decreasing $g$ implies that the monopole contains a larger region of true vacuum at its core which for sufficiently large size to become classically unstable.

From equations (5) & (15) the monopole induced rate will dominate that of tunnelling from the homogenous false vacuum for a monopole abundance which satisfies

$$n_m \gtrsim v^3 e^{B_{\mathrm{mb}} - B_{\mathrm{fvb}}}. \tag{32}$$

This can be written in terms of the ratio of the monopole number density per Hubble volume as

$$\frac{n_m}{H^3} > e^{B_{\mathrm{mb}} - B_{\mathrm{fvb}}} \left( \frac{\pi^2 g_*}{30} \right)^{-3/2} \left( \frac{v M_P}{T^2} \right)^3, \tag{33}$$

assuming a radiation dominated universe with $g_*$ degrees of freedom. The difference in the bounce actions for each of the points shown in figures 6 falls in the range

$$
\begin{aligned}
B_{\mathrm{mb}} - B_{\mathrm{fvb}} &\in [-360, -150], \\
\implies e^{B_{\mathrm{mb}} - B_{\mathrm{fvb}}} &\in [4 \times 10^{-157}, 8 \times 10^{-66}].
\end{aligned}
\tag{34}
$$

If we take the monopoles to be produced via the Kibble-Zurek mechanism (discussed further in section 4.1) when the universe has a temperature $T_* \sim v$ then we would expect of order 1 monopole per Hubble volume at this temperature [48, 49]. As the universe cools the monopole abundance scales as:

$$n_{m,\mathrm{kibble}} = H_*^3 \left( \frac{T}{T_*} \right)^3 \simeq H^3 \left( \frac{v}{T} \right)^3, \tag{35}$$

for constant $g_*$. In this case we find that the monopole-catalysed process will dominate the false vacuum phase transition rate for:

$$\frac{T}{M_P} \gtrsim 1.7 \times 10^{-23} \left( \frac{e^{B_{\mathrm{mb}} - B_{\mathrm{fvb}}}}{10^{-66}} \right)^{1/3} . \tag{36}$$

For any viable BSM phase transition monopoles produced through the Kibble-Zurek mechanism will give the dominant contribution to the phase transition rate, within all of our parameter space.

We can also consider the stability of the present-day vacuum in the presence of a monopole population which makes up a fraction $\Omega_m$ of the total energy density of the universe. The condition that the monopole induced rate dominates the false vacuum rate is that:

$$\Omega_m > 1.3 \times 10^{-23} g^3 \left( \frac{M_m}{\mathrm{GeV}} \right)^4 \left( \frac{e^{B_{\mathrm{mb}} - B_{\mathrm{fvb}}}}{10^{-66}} \right), \tag{37}$$

where $g$ is the gauge coupling and we have used the approximate relation $M_m = 4\pi v/g$. For example, a single monopole in our Hubble patch $\Omega_m \sim M_m H / M_P^2$ dominates the decay if $B_{\mathrm{mb}} - B_{\mathrm{fvb}} \lesssim -300$ for monopole masses of order $M_m \sim$ TeV and $B_{\mathrm{mb}} - B_{\mathrm{fvb}} \lesssim -386$ for monopole masses of order $M_m \sim M_{\mathrm{GUT}} \sim 10^{16}$ GeV.

For monopoles with magnetic charge under $U(1)_{\mathrm{em}}$, there is a bound on the monopole abundance [50, 51]

$$\Omega_m < 1.3 \times 10^{-16} \left( \frac{M_m}{\mathrm{GeV}} \right) \left( \frac{\beta_m}{10^{-3}} \right)^{-1} , \tag{38}$$

assuming a constant monopole density throughout the universe, where $\beta_m$ is the velocity of monopoles in the galaxy. The critical value of the bounce action required for electromagnetic monopoles to play a significant role in catalysing vacuum decay today depends on their mass. For monopoles with mass $M_m \sim$ TeV conditions (37) & (38) can be satisfied for $B_{\mathrm{mb}} - B_{\mathrm{fvb}} \lesssim -163$, while if $M_m \sim M_{\mathrm{GUT}}$ then both requirements are satisfied for $B_{\mathrm{mb}} - B_{\mathrm{fvb}} \lesssim -258$.

# 4 Phenomenological Implications

The results of section 3 highlight the significant enhancement in the false vacuum decay rate due to the presence of magnetic monopoles in a simple model. In this section we present some examples illustrating how our calculation of monopole-induced tunnelling could apply in more general models of BSM physics.

## 4.1 Metastable monopole production

A common method by which monopoles are produced in the early universe is via the Kibble-Zurek mechanism [48, 49]. This occurs when the universe undergoes a phase transition to the symmetry-broken phase, where the symmetry breaking pattern allows monopole solutions. Points separated by distances larger than the correlation length exit the phase transition at different points on the vacuum manifold, resulting in twisted field configurations which relax to magnetic monopole solutions.

This can be a viable production for metastable monopoles in theories with two-step phase transitions where the $SU(2)$ gauge group is unbroken at high temperatures, is then broken in the first phase transition where monopoles are produced via the Kibble-Zurek mechanism.

In the second phase transition an $SU(2)$ preserving vacuum state becomes the lowest energy state making the monopole metastable. An example of this was considered in ref.'s [15, 16] in the context of an $SU(5)$ GUT breaking to $SU(4) \times U(1)$ before breaking to the SM. Another related possibility is that during the cosmological evolution after the first phase transition, the symmetry-breaking vacuum (which is initially the preferred vacuum) becomes metastable due to thermal corrections to the potential. The monopole abundance in either case is $\mathcal{O}\left(\xi^{-3}\right)$, where $\xi$ is the correlation length of the phase transition, which is Hubble scale for a first order transition but can be significantly smaller than Hubble for second-order transitions [52].

The Kibble-Zurek mechanism requires the early phase of the universe to be one in which the $SU(2)$ symmetry is restored, however we can consider alternate production mechanisms when this is not the case. Examples of this are monopoles produced by Hawking evaporation of primordial black holes [53], Schwinger pair-production in magnetic fields [54] or in high-energy particle collisions [55]. As the monopoles can be classically unstable when the false vacuum tunnelling process is exponentially suppressed, even if magnetic monopoles are very rarely produced through particle collisions or black hole evaporation, these processes may still trigger false vacuum decay. Another interesting production mode that has recently been considered is through the Kibble-Zurek mechanism operating in the thermal plasma surrounding an evaporating black hole [56].

## 4.2 GUT phase transitions

Although we considered the simplest possible case of a scalar potential with true vacuum at $\phi = 0$, monopole decay may also be relevant for theories where the field values at the monopole core may not be at the precise true vacuum, but simply in the basin of attraction of the true vacuum. This can have implications for the lifetime of the current vacuum state of the universe. If the UV completion of the SM is described by a GUT, then it is possible that the universe is in a metastable phase described by

$$\langle \Phi_{\mathrm{GUT}} \rangle = M_{\mathrm{GUT}} \,, \tag{39}$$

where $\Phi_{\mathrm{GUT}}$ is a scalar field which spontaneously breaks the GUT symmetry. At the core of a GUT monopole, $\langle \Phi_{\mathrm{GUT}} \rangle = 0$ and if this field value is in the basin of attraction of a new, lower minimum in the scalar potential then GUT monopoles are metastable. One way that this could occur is through interplay with the Higgs potential. As is well known, current experimental measurements of the top Yukawa indicate that the Higgs quartic turns negative at a large field value $h > 10^{10}$ GeV [11]. This contribution can help ensure that a second minimum with a lower vacuum energy exists, however, the details are model dependent. Given that this decay has evidently not occurred at any point in our past lightcone, this can be used to put bounds on the abundance of GUT monopoles, although we leave a detailed analysis of this possibility for future work.

## 4.3 Gravitational waves

The gravitational wave signal from a first order phase transition can also be significantly altered if it is catalysed by a population of magnetic monopoles. The fraction of the total energy emitted as gravitational waves scales and the peak frequency of the resulting spectrum scale as [57–62]

$$\frac{E_{\mathrm{GW}}}{E_{\mathrm{vac}}} \propto \left(\frac{H}{\beta}\right)^2 \,,$$
$$\omega_{\mathrm{max}} \propto \beta \,, \tag{40}$$

where $\beta^{-1}$ is the time the nucleated bubbles evolve before colliding and $H$ is the Hubble rate at the time of the phase transition. As the bubbles rapidly accelerate until they are moving at the speed of light, $\beta^{-1}$ is given by the average separation of bubbles when they are nucleated.[4]

For the false vacuum process $\beta^{-1}$ is determined solely by the bounce action and how it varies with time, as bubbles are equally likely to be nucleated at any point in space. If the magnetic monopoles are homogeneously distributed with number density larger than one per Hubble volume the monopole catalysed process behaves in the same way as the false vacuum transition, although the rate may differ. If the phase transition completes more rapidly due to the presence of monopoles the amplitude of the gravitational wave signal will be decreased and shifted to higher frequencies. However, if the population of monopoles is not uniformly distributed on Hubble scales then $\beta^{-1}$ will instead be set by the average separation of the monopoles. In this case there is also the possibility that there may be multiple scales on which bubbles of different sizes collide, leading to a spectrum with multiple peaks of differing intensity. Furthermore, the topology of the bubble walls may introduce additional interactions between the walls or between the walls and the thermal plasma which may modify the dynamics of the wall.

A further interesting possibility is that the distribution of monopoles could lead to an observable anisotropy in the gravitational wave spectrum. In ref. [63] it was shown that upcoming gravitational wave experiments are potentially sensitive to spatial anisotropies in the gravitational wave signal from a first order phase transition. It was also shown that for a generic first order phase transition that this spectrum should be proportional to the temperature anisotropy,

$$\frac{\delta\rho_{GW}(\theta,\phi)}{\rho_{GW}} \propto \frac{\delta T(\theta,\phi)}{T}, \tag{41}$$

and therefore gives a second copy of the CMB (without the effects of Silk damping and baryon acoustic oscillations). If the phase transition was instead catalysed by a population of magnetic monopoles whose abundance has a spatial dependence that is not correlated with temperature this could lead to a spatial variation in the gravitational wave signal that is not aligned with the CMB. One example of how this may occur is if primordial magnetic fields bias the Kibble-Zurek mechanism to produce more monopoles in some regions of space than others. The gravitational wave anisotropy would then additionally track the anisotropy in the monopole number density

$$\frac{\delta\rho_{GW}(\theta,\phi)}{\rho_{GW}} \sim \frac{\delta n_m(\theta,\phi)}{n_m}, \tag{42}$$

which in turn depends on the magnetic field strength at different points in space.

## 5 Conclusion

First order phase transitions are expected to play a major role in the cosmology in many models of BSM physics. In this work, we have shown that the dynamics of these phase transitions may be dramatically altered by a population of metastable magnetic monopoles which act to catalyse the phase transition. The requirement for monopoles to be metastable is that the field values at the monopole core are close to the true vacuum of the potential. These monopoles may be produced via the Kibble mechanism in a two-step phase transition, in high energy particle collisions or be emitted during black hole evaporation. In the presence of these defects

---

[4]Here we are ignoring interactions of the bubble walls with the thermal plasma.

the phase transition rate can be exponentially enhanced, and the expected gravitational wave signal caused by the phase transition may also be significantly modified. This can also have implications for the stability of the vacuum if some population of monopoles exists in the current universe.

The idea of defect-catalysed phase transitions has wider applicability than the case considered in this work. While we have studied zero temperature monopole decays in this paper, decays assisted by thermal effects or high energy collisions are also possible. These processes can produce excitations of the metastable monopoles over the critical barrier that are classically unstable and trigger the false vacuum phase transitions. Furthermore, in a given UV completion SM or BSM particles may have some characteristic size and be metastable against expansion. In this case they can play the role of catalysing defects similar to the monopole case considered here.

The algorithm presented in section 3 may have further applications beyond defect-catalysed phase transitions. The algorithm was constructed to determine solutions to the Euclidean field equations which depend independently on $r$ and $\tau$. Most calculations of the bounce action for cosmological phase transitions are done assuming the dominant bounce solution depends solely on $\rho$ or on $r$, which is only true in the $T = 0$ or $T \to \infty$ limit, respectively. The procedure described in section 3 can be used to calculate the bounce action at finite temperature in full generality.

There are many future directions to study, such as the gravitational wave signal, production mechanisms and stability of the SM vacuum in GUT theories. The boring monopole turns out to have many interesting consequences.

## Acknowledgements

We would like to thank Anson Hook and Junwu Huang for their comments on a draft of the manuscript. We are grateful to John March-Russell and John Wheater for useful conversations.

**Funding information** PA is supported by the STFC under Grant No. ST/T000864/1. MN is funded by a joint Clarendon and Sloane-Robinson scholarship from Oxford University and Keble college.

## A  Finding monopole profiles

In this appendix we describe the modified shooting algorithm used to derive monopole profiles. Considering equations (18) and the relevant boundary conditions it can be shown that the monopole profiles admit a Taylor expansion at small $s$

$$h_m(s) \simeq h_0 s, \qquad\qquad u_m(s) \simeq 1 - u_0 s^2, \qquad\qquad \text{(A.1)}$$

and at large $s$ must asymptote to the solutions

$$h_m(s) = 1 - c_h e^{-a_h s}, \qquad\qquad u_m(s) = c_u e^{-s}, \qquad\qquad \text{(A.2)}$$

where $a_h = \lambda g^{-2} \frac{\partial^2 V(h)}{\partial h^2}|_{h=1}$ and $h_0, u_0, c_h, c_u$ are undetermined parameters. To solve equations (18) over the full range of $s$ values we start with an initial guess for each of the four parameters and solve the equations of motion in two seperate regimes. Over a range $s \in [10^{-2}, 1]$ we solve the equations as an initial value problem with initial conditions at $s = 10^{-2}$ derived

from the Taylor expansions (A.1) for the chosen values of $h_0, u_0$, giving us the 'forward' profiles $(h_+, u_+)$ defined on $s \in [10^{-2}, 1]$. We then separately solve the equations for the range $s \in [1, 10]$ starting at $s = 10$. Working backwards, we use the initial conditions defined by the asymptotic expansion (A.2) for the chosen values of $c_h, c_u$, giving us 'backward' profiles $(h_-, u_-)$ defined for $s \in [1, 10]$.

The next step is to check how well the solutions match at $s = 1$ by calculating a test function

$$\vec{T}(\vec{p}) = \vec{y}_+(1) - \vec{y}_-(1), \tag{A.3}$$

where $\vec{y}_+, \vec{y}_-$ are given by the vector

$$\vec{y}(s) = (h(s), h'(s), u(s), u'(s))^T, \tag{A.4}$$

defined for the forward and backward solutions respectively. Here we are considering $\vec{T}$ as a function of the parameters $\vec{p} \equiv (h_0, u_0, c_h, c_u)$ which returns a four vector. The solution is then given by the set of parameters which give $\vec{T} = 0$. If our given choice of $\vec{p}$ does not give a solution we update the set of parameters using Newton's method, varying each parameter by an amount

$$\Delta \vec{p} = -J^{-1} \vec{T}(\vec{p}), \tag{A.5}$$

where $J$ is the Jacobian matrix with components

$$J_{ij} = \frac{\partial T_i}{\partial p_j}. \tag{A.6}$$

To determine the Jacobian we write the system of differential equations in the form

$$\frac{\partial}{\partial s} \vec{y} = \vec{f}(s, \vec{y}). \tag{A.7}$$

Taking the partial derivative with respect to the components of $\vec{p}$ gives the following equation

$$\frac{\partial}{\partial s} \left( \frac{\partial y_i}{\partial p_j} \right) = \frac{\partial f_i(s, \vec{y})}{\partial y_k} \frac{\partial y_k}{\partial p_j}, \tag{A.8}$$

which can be solved concurrently with the equations of motion (18) to determine the matrix

$$z_{ij}(s) \equiv \frac{\partial y_i}{\partial p_j}. \tag{A.9}$$

The Jacobian $J$ is given by $J_{ij} = (z_+)_{ij} - (z_-)_{ij}$ evaluated at the point $s = 1$, where $z_+, z_-$ are the above matrices solved for the forward and backward solutions.

Due to the difficulty in solving the non-linear equations of motion it was useful to obtain a good initial guess by solving the monopole solutions for a mild choice of $\epsilon, \lambda, g \lesssim 1$ first. To construct solutions away from this set of values space we took this solution, then varied the parameters $\epsilon, \lambda, g$ incrementally away from the initial choice of parameters, using the vector $\vec{p}$ obtained from the previous solution as the initial guess to begin the algorithm for the next set of parameters. It was also necessary to choose a smaller step size when updating the parameters, modifying equation (28) to include a parameter $\xi$

$$\Delta \vec{p} = -\xi J^{-1} T(h_0, u_0, c_h, c_u). \tag{A.10}$$

For a mild set of parameters, $\xi = 1$ was sufficient for the algorithm to converge, but for less well-behaved parameters reducing the step size was necessary to ensure convergence. To implement this, we began with a step size $\xi = 1$, and whenever the algorithm began to diverge, halved the step size and started again.

# B  Discretisation

While our discussion of the procedure in section 3.2 considered the functions $h_\alpha^n, u_\alpha^n$ to be continuous in each of the variables $\tau, s$ and $\alpha$, in order to implement the MPT algorithm it is necessary to discretise the problem. To do this, we first construct a lattice of points $(s_i, \tau_j, \alpha_k)$, where we take the points $s_i$ to be logarithmically spaced. The points were chosen to be in the intervals

$$10^{-2} \le s \le 2S, \quad 0 \le \tau \le 4T, \quad 0 \le \alpha \le 1, \tag{B.1}$$

for $T, S$ taken from the super-critical solution given in equation (25). The functions are then written as tensors $h_{ijk}, u_{ijk}$ such that

$$h_{ijk} = h_{\alpha_k}(\tau_j, s_i), \qquad\qquad u_{ijk} = u_{\alpha_k}(\tau_j, s_i). \tag{B.2}$$

In the discretised theory the Euclidean action becomes

$$S_E[h_{ijk}, u_{ijk}] = \frac{4\pi}{g^2} \sum_{ij} s_i^2 ds_i d\tau \left\{ \frac{1}{2} \left( \dot{h}_{ijk}^2 + h'^2_{ijk} \right) + \frac{1}{g^2} \left( \dot{u}_{ijk}^2 + u'^2_{ijk} \right) \right.$$
$$\left. + \frac{h_{ijk}^2 u_{ijk}^2}{s^2} + \frac{1}{2} \left( u_{ijk}^2 - 1 \right)^2 + \frac{\lambda}{g^2} V_h(h_{ijk}) \right\}, \tag{B.3}$$

where $d\tau$ is the lattice spacing in the $\tau$ direction and $ds_i$ the lattice spacing in $s$, which we note has an index because the lattice in the $s$-direction is logarithmically spaced. The derivatives are approximated as finite differences and we use the boundary conditions to compute the derivatives at the endpoints. For example, approximating $\dot{h}_{ijk}$ as a finite difference at a given point $(\tau_j, s_i)$ gives

$$\dot{h}_{ijk} = \frac{h_{i,j+1,k} - h_{i,j-1,k}}{2d\tau}. \tag{B.4}$$

At the endpoint ($j$ such that $\tau_j = 4T$) we let $h_{i,j+1,k} = h_m(s_i)$, enforcing the boundary condition $h(\tau, s) \to h_m(s)$ at large $\tau$. For the boundary condition at $\tau = 0$ we use the reflection symmetry $\tau \to -\tau$, at large $s$ we require that $h \to 1, u \to 0$ and at small $s$ we use a Taylor expansion.

The discretised action is used to calculate equation (26) for each value of $k$ and determine the value $\bar{k}$ for which $I_{\alpha_{\bar{k}}}^n$ is a maximum at a given step $n$. The discretised form of equations (29) is then

$$\left( \frac{\delta \bar{I}}{\delta h_{\tilde{\alpha}}^n} \right)_{ij} = -\frac{8\pi s_i^2}{g^2} \left( \ddot{h}_{ij\bar{k}} + h''_{ij\bar{k}} + \frac{2}{s_i} h'_{ij\bar{k}} - \frac{2h_{ij\bar{k}} u_{ij\bar{k}}^2}{s_i^2} - \lambda V_h'(h_{ij\bar{k}}) \right), \tag{B.5}$$

$$\left( \frac{\delta \bar{I}}{\delta u_{\tilde{\alpha}}^n} \right)_{ij} = -\frac{16\pi}{g^2} \left( \ddot{u}_{ij\bar{k}} + u''_{ij\bar{k}} - \frac{u_{ij\bar{k}}(u_{ij\bar{k}}^2 - 1)}{s_i^2} - h_{ij\bar{k}}^2 u_{ij\bar{k}} \right), \tag{B.6}$$

where derivatives are again approximated using finite difference methods as described above. The profiles are then updated according to the gradient descent algorithm as in equation (28)

$$h_{ijk}^{n+1} = h_{ijk}^n - \beta_n F(\alpha_k, \alpha_{\bar{k}}) \left( \frac{\delta \bar{I}}{\delta h_{\tilde{\alpha}}^n} \right)_{ij},$$
$$u_{ijk}^{n+1} = u_{ijk}^n - \beta_n F(\alpha_k, \alpha_{\bar{k}}) \left( \frac{\delta \bar{I}}{\delta u_{\tilde{\alpha}}^n} \right)_{ij}. \tag{B.7}$$

The step size $\beta_n$ we choose depends on the value of $\bar{k}$ which maximises $I^n_{\alpha_k}$ and whether the value found at step $n$ differs from the value at the previous step $n-1$. To calculate the step size we let $\vec{X}^n_k$ be the vector constructed from combining the components of $(h^n_{ijk}, u^n_{ijk})$ and $\delta \vec{X}^n_k = \vec{X}^n_k - \vec{X}^{n-1}_k$. The step size we use is [64]

$$\beta_n = \frac{|\left(\vec{X}^n_{\bar{k}} - \vec{X}^{n-1}_{\bar{k}}\right) \cdot \left(\delta\vec{X}^n_{\bar{k}} - \delta\vec{X}^{n-1}_{\bar{k}}\right)|}{\|\delta\vec{X}^n_{\bar{k}} - \delta\vec{X}^{n-1}_{\bar{k}}\|} \, . \tag{B.8}$$

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
