# Peer review of "The Boring Monopole"

_SciPost Physics, doi:SciPost Phys. 13, 049 (2022)_

## Round 2 · Referee Report · Anonymous (Referee 2) · 2022-5-24

Strengths

The mountain pass theorem and algorithms are clearly explained.

Weaknesses

The main statement is misleading.
The scenario is specific to the model they consider but it not clearly explained in the abstract and conclusion.

Report

The authors study false vacuum decay catalyzed by monopoles in a model where the monopole is metastable and its center corresponds to the symmetry-preserved true-vacuum state. This possibility was considered in [15, 16, 17, 18] in a similar setup, whereas the present paper provides a method to calculate the catalyzed decay rate without the thin-wall approximation. The method they used seems to be important for the study of vacuum decay even in a different context. I would ask the authors to address the following questions and comments before recommending the publication.

Which part of this work is original about the numerical method? It is not clear for me if the mountain pass algorithm is new or had been proposed in the literature.

The statement of the paper, especially in the abstract and conclusion, is misleading. The authors consider a very specific setup, where the monopole is metastable and its center corresponds to the symmetry-preserved true-vacuum state. This seems to be constructed in such a way that the monopole can catalyze the vacuum decay. Moreover, the production of such metastable monopoles require a complicated cosmological history as the authors mention in Sec. 4.1. The authors should clarify these assumptions in the abstract and conclusion so that readers do not misunderstand that any monopoles catalyze the vacuum decay.

The statement about the lifetime of the Standard Model vacuum is also misleading. The authors discuss the effect of catalyzed decay on the metastable EW vacuum. They state that the catalyzed decay by monopole is problematic because it leads to the Universe filling with $<H> = v_h$ and $<\Phi_{GUT}> = 0$. I agree that this is catastrophic. However, I think this issue is not related to the metastable EW vacuum. Since the GUT scale is much higher than $v_h$, the effect of Higgs potential to the decay rate should be negligible. No matter how the Higgs field changes by the catalyzed decay, the phase transition from $<\Phi_{GUT}> = M_{GUT}$ to $<\Phi_{GUT}> \sim 0$ is anyway problematic because the gauge structure of SM changes. It seems to me that the catalyzed vacuum decay by the monopole has nothing to do with the issue of the metastable EW vacuum.

Requested changes

Clarify if the method proposed in Sec. 3.2 is original or not, and cite appropriate references if any.
Add comments on the assumptions in the abstract and conclusion.
Revise the statement about the lifetime of the Standard Model vacuum.

  • validity: ok
  • significance: good
  • originality: good
  • clarity: ok
  • formatting: excellent
  • grammar: excellent

Author:  Michael Nee  on 2022-06-29  [id 2616]

(in reply to Report 2 on 2022-05-24)

We thank the referee for their comments and a careful reading of the manuscript. We have made changes to the abstract and conclusion to better qualify our main statement. We list here the explicit changes made in response to the comments made by the referee: 1. The mountain pass theorem is well-known in the mathematical literature, cited in our paper as [47], and given its basic nature probably appears in a different guise in a number of applications. As far as we are aware, the application to tunnelling calculations in false vacuum decay is original. We have added a comment outlining this on page 10, paragraph 2. 2. We have added added qualifications to our statements in the abstract and in the first paragraph of the conclusion. We agree that our results do not apply to general monopoles which can be stable. However, we wish to emphasize that the particular example is only chosen for concreteness, and the idea that defect-catalyzed decays might vastly dominate over homogenous decay is much more general (such as the stability of electric fields in vacuum vs. in presence of a dilute gas). 3. We have clarified our statements in section 4.2 regarding the lifetime of the electroweak vacuum. We agree that decays catalysed by GUT monopoles do not necessarily involve Higgs dynamics. The section aims to highlight a realistic visible sector example with metastable monopoles. It will be interesting to explore the interplay between the Higgs quartic running to negative values and the dynamics of GUT breaking in a concrete model in the future.

---

## Round 2 · Referee Report · Anonymous (Referee 1) · 2022-5-24

Strengths

New application of mountain path theorem to vacuum decay

Weaknesses

The arguments in section4 are vague, and there is no quantitative results.

There is not clear explanation for a negative mode around bounce solution.

Report

In this paper, the authors studied the enhanced decay rate of a false vacuum by the effect of monopole catalysis. Especially, with the help of the mountain pass theorem (MPT), they investigated an algorithm for numerical estimation for the decay rate. Applying MPT method to false vacuum decay is quite interesting, and it can be discussed in a large number of other models.

However, before considering publication, I would like the authors to clarify the consistency of a negative mode around the bounce solution and the MPT method: MPT turns solving equations into a minimization problem. However, bounce solution, that is need to estimate decay rate, is not the absolute minimum of the Euclidean action but a saddle point, which makes numerical calculation involved, in general. The same complexity should exist in this mountain pass method. This is essential when one numerically estimates the rate by using variation paths because some paths can include the negative mode around the classical solution, which leads to lower values of action.

Also, one of the virtue of MPT is independence on an initial profile. However, if the initial profile includes the negative mode of the bounce solution, one would get lower value than the precise bounce action. How can the authors justify this in the calculation in section 3? The results shown in figure 5 could be lower than the precise value.

In this way, as for this negative mode, I think that the argument shown in section 3, is not enough for the reader to understand them. I think that the authors should add some comments on this issue.

Requested changes

Add comments on relation between the negative mode and MPT.

  • validity: ok
  • significance: good
  • originality: good
  • clarity: good
  • formatting: good
  • grammar: excellent

Author:  Michael Nee  on 2022-06-29  [id 2617]

(in reply to Report 1 on 2022-05-24)

We thank the referee for their comments and a careful reading of the manuscript. We include a response to the comment made by the referee and list the changes made in response in the manuscript.

"The arguments in section 4 are vague, and there is no quantitative results."

We agree with the referee that section 4 is more qualitative than the rest of the paper. In this paper, we wished to emphasize the mechanism of catalyzed vacuum decay, and the mountain pass theorem as a calculational tool for the bounce action. The examples presented in section 4 aim to connect this quantum field theoretic aspect to realistic phenomenological models, the detailed quantitative study of each of which will potentially be a paper itself. Indeed, we think these are some of the more promising follow-up directions to pursue in the future.

"Add comments on relation between the negative mode and MPT.
The mountain pass theorem is designed to specifically deal with the instability caused by the existence of a negative mode and the min/max procedure ensures numerical convergence. The initial profiles may well contain the negative mode, however, this does not give an artificially lower value of the bounce action. At each iteration the MPT chooses a path that is an interpolation between two profiles on either side of the tunneling barrier, and selects the maximum action point along this path. This ensures that the negative mode is not lowering the action. In fact, the action at this maximum point will always be bigger than the action at the saddle. Next, a variation of paths is carried out, and when the path goes through the "mountain pass", the maximum along the path is precisely on the saddle. We have added comments on page 11, paragraph 1 to better explain this in the manuscript.

---

## Round 3 · Referee Report · Anonymous (Referee 2) · 2022-7-15

Report

The authors have addressed the questions and comments mentioned in the referee report. I recommend publication of this paper in SciPost.

---

## Round 3 · Referee Report · Anonymous (Referee 1) · 2022-8-4

Report

The authors addressed the issues raised in the report. So, I recommend publishing this paper in SciPost.

---

## Editorial Decision

published